# Information bottleneck theory of high-dimensional regression: relevancy, efficiency and optimality

**Vudtiwat Ngampruetikorn,**[*] **David J. Schwab**
Initiative for the Theoretical Sciences, The Graduate Center, CUNY
[*]vngampruetikorn@gc.cuny.edu

## Abstract

Avoiding overfitting is a central challenge in machine learning, yet many large neural networks readily achieve zero training loss. This puzzling contradiction necessitates new approaches to the study of overfitting. Here we quantify overfitting via residual information, defined as the bits in fitted models that encode noise in training data. Information efficient learning algorithms minimize residual information while maximizing the relevant bits, which are predictive of the unknown generative models. We solve this optimization to obtain the information content of optimal algorithms for a linear regression problem and compare it to that of randomized ridge regression. Our results demonstrate the fundamental trade-off between residual and relevant information and characterize the relative information efficiency of randomized regression with respect to optimal algorithms. Finally, using results from random matrix theory, we reveal the information complexity of learning a linear map in high dimensions and unveil information-theoretic analogs of double and multiple descent phenomena.

## 1 Information bottleneck

Conventional wisdom identifies overfitting as being detrimental to generalization performance, yet modern machine learning is dominated by models that perfectly fit training data. Recent attempts to resolve this dilemma have offered much needed insight into the generalization properties of perfectly fitted models [1, 2]. However investigations of overfitting beyond generalization error have received less attention. In this work we present a quantitative analysis of overfitting based on information theory and, in particular, the information bottleneck (IB) method [3].

The essence of learning is the ability to find useful and generalizable representations of training data. An example of such a representation is a fitted model which may capture statistical correlations between two variables (regression and pattern recognition) or the relative likelihood of random variables (density estimation). While what makes a representation useful is problem specific, a good model generalizes well—that is, it is consistent with test data even though they are not used at training.

Achieving good generalization requires information about the unknown data generating process. Maximizing this information is an intuitive strategy, yet extracting too many bits from the training data hurts generalization [4, 5]. This fundamental trade-off underpins the IB principle, which formalizes the notion of a maximally efficient representation as an optimization problem [3][1]

$$\min_{Q_{T|S}} I(S;T \mid W) - (\gamma - 1)I(T;W). \tag{1}$$

Here $W$ denotes the data generating process. The conditional distribution $Q_{T|S}$ denotes a learning algorithm which defines a stochastic mapping from the training data $S$ to the hypothesis or fitted

---

[1]Note that this minimization is identical to that of the original IB method since $I(S;T|W) = I(S;T) - I(T;W)$ under the Markov constraint $T \leftrightarrow S \leftrightarrow W$.

36th Conference on Neural Information Processing Systems (NeurIPS 2022).

model $T$ (Fig 1a). The *relevant information*, $I(T;W)$, is the bits in $T$ that are informative of the generative model $W$. On the other hand, the *residual information*, $I(S;T \mid W)$, is the bits in $T$ that are specific to each realization of the training data $S$ and thus are not informative of $W$. In other words the residual bits measure the degree of overfitting. The parameter $\gamma$ controls the trade-off between these two informations.

The IB method has found success in a diverse array of applications, from neural coding [6, 7], developmental biology [8] and statistical physics [9–11] to clustering [12], deep learning [13–15] and reinforcement learning [16].

Indeed the IB principle has emerged as a potential candidate for a unifying framework for understanding learning phenomena [15, 17–19] and a number of recent works have explored deep connections between information-theoretic quantities and generalization properties [4, 5, 20–28]. However a direct application of information theory to practical learning algorithms is often limited by the difficulty in estimating information, especially in high dimensions. While recent advances in characterizing variational bounds of mutual information have enabled a great deal of scalable, information-theory inspired learning methods [13, 29, 30], these bounds are generally loose and may not reflect the true behaviors of information.

To this end we consider a tractable problem of learning a linear map. We show that the level of overfitting, as measured by the encoded residual bits, is nonmonotonic in sample size, exhibiting a maximum near the crossover between under- and overparametrized regimes. We also demonstrate that additional maxima can develop under anisotropic covariates. As the residual information bounds the generalization gap [4, 5], its nonmonotonicity can be viewed as an information-theoretic analog of (sample-wise) multiple descent—the existence of disjoint regions in which more data hurt generalization (see, e.g., Refs [31, 32]). Using an IB optimal representation as a baseline, we show that the information efficiency of a randomized least squares regression estimator exhibits sample-wise non-monotonicity with a minimum near the residual information peak. Finally we discuss how redundant coding of relevant information in the data gives rise to the nonmonotonicity of the encoded residual bits and how additional maxima emerge from covariate anisotropy (Sec 4).

## 1.1 Generative model

**Linear map**—We consider training data of $N$ iid samples, $S = \{(x_1, y_1), \ldots, (x_N, y_N)\}$, each of which is a pair of $P$ dimensional input vector $x_i \in \mathbb{R}^P$ and scalar response $y_i \in \mathbb{R}$ for $i \in \{1, \ldots, N\}$. We assume a linear relation between the input and response,

$$y_i = W \cdot x_i + \epsilon_i \quad \text{and} \quad \epsilon_i \sim N(0, \sigma^2), \tag{2}$$

where $W \in \mathbb{R}^P$ denotes the unknown linear map and $\epsilon_i$ a scalar Gaussian noise with mean zero and variance $\sigma^2$. In other words the responses and the inputs are related via a Gaussian channel

$$Y \mid X, W \sim N(X^\mathsf{T} W, \sigma^2 I_N), \tag{3}$$

where we define $Y = (y_1, \ldots, y_N)^\mathsf{T} \in \mathbb{R}^N$ and $X = (x_1, \ldots, x_N) \in \mathbb{R}^{P \times N}$.

**Fixed design**—We adopt the fixed design setting in which the inputs (design matrix) $X$ are deterministic and only the responses $Y$ are random variables (see, e.g., Ref [33, Ch 3]). As a result, the mutual information between the training data $S$ and any random variable $A$ is given by $I(A;S) = I(A;X,Y) = I(A;Y)$. In the following analyses, we use $S$ and $Y$ interchangeably.

**Random effects**—In addition we work in the random effects setting (see, e.g., [34, 35] for recent applications of this setting) in which the true regression parameter $W$ is a Gaussian vector,

$$W \sim N(0, \tfrac{\omega^2}{P} I_P). \tag{4}$$

Here we define the covariance such that the signal strength, $\mathbb{E} \|W\|^2 = \omega^2$, is independent of $P$.

## 1.2 Information optimal algorithm

The data generating process above results in training data $Y$ and true parameters $W$ that are Gaussian correlated (under the fixed design setting). In this case the IB optimization—minimizing residual information $I(T;Y \mid W)$ while maximizing relevant information $I(T;W)$—admits an exact solution [36],

characterized by the eigenmodes of the normalized regression matrix,

$$\Sigma_{Y|W}\Sigma_Y^{-1} = \left(I_N + \frac{1}{\lambda^*}\frac{X^\mathsf{T}X}{N}\right)^{-1} \quad \text{with} \quad \lambda^* \equiv \frac{P}{N}\frac{\sigma^2}{\omega^2}, \tag{5}$$

where $\lambda^*$ denotes the scaled noise-to-signal ratio. The relevant and residual informations of an optimal representation $\tilde{T}$ read [36]

$$I(\tilde{T};W) = \frac{1}{2}\sum_{i=1}^{N} \max(0,\ \ln((1-\gamma^{-1})/\nu_i)) \tag{6}$$

$$I(\tilde{T};Y\mid W) = \frac{1}{2}\sum_{i=1}^{N} \max(0,\ \ln(\gamma(1-\nu_i))), \tag{7}$$

where $\nu_i$ denote the eigenvalues of $\Sigma_{Y|W}\Sigma_Y^{-1}$ and $\gamma$ parametrizes the IB trade-off[2] [see Eq (1)]. In our setting it is convenient to recast the summations above as integrals (see Appendix A for derivation),

$$I(\tilde{T};W) = \frac{P}{2}\int_{\psi>\psi_c} dF^{\Psi}(\psi)\ \ln\left(1+\frac{\psi-\psi_c}{\psi_c+\lambda^*}\right) \tag{8}$$

$$I(\tilde{T};Y\mid W) = \frac{P}{2}\int_{\psi>\psi_c} dF^{\Psi}(\psi)\ \ln(\psi/\psi_c) - I(\tilde{T};W), \tag{9}$$

where $\Psi \equiv XX^\mathsf{T}/N$ and $F^{\Psi}$ denote the empirical covariance and its cumulative spectral distribution, respectively.[3] In addition we introduce the parameter $\psi_c = \lambda^*/(\gamma-1)$ which controls the number and the weights of eigenmodes used in constructing the optimal representation $\tilde{T}$. In the limit $\psi_c \to 0^+$, the residual information diverges logarithmically (Fig 1d) and the relevant information converges to the available relevant information in the data (Fig 1c),

$$I(\tilde{T};W) \overset{\psi_c\to 0^+}{\to} I(Y;W) = \frac{P}{2}\int_{\psi>0} dF^{\Psi}(\psi)\ \ln(1+\psi/\lambda^*). \tag{10}$$

Increasing $\psi_c$ from zero decreases both residual and relevant informations, tracing out the optimal frontier until the lower spectral cutoff $\psi_c$ reaches the upper spectral edge at which both informations vanish (Fig 1b) and beyond which no informative solution exists [36–38].

## 1.3 Information efficiency

The exact characterization of the IB frontier provides a useful benchmark for information-theoretic analyses of learning algorithms, not least because it allows a precise definition of information efficiency. That is, we can now ask how many more residual bits a given algorithm needs to encode, compared to the IB optimal algorithm, in order to achieve the same level of relevant information. Here we define the information efficiency $\eta_\mu$ as the ratio between the residual bits encoded in the outputs of the IB optimal algorithm and the algorithm of interest—$\tilde{T}$ and $T$, respectively—at some fixed relevance level $\mu$, i.e.,

$$\eta_\mu \equiv \frac{I(\tilde{T};Y\mid W)}{I(T;Y\mid W)} \quad \text{subject to} \quad \mu = \frac{I(\tilde{T};W)}{I(Y;W)} = \frac{I(T;W)}{I(Y;W)}. \tag{11}$$

Since the optimal representation minimizes residual bits at fixed $\mu$ (Fig 1a), the information efficiency ranges from zero to one, $0 \le \eta_\mu \le 1$. In addition we have $0 < \mu \le 1$, resulting from the data processing inequality $I(T;W) \le I(Y;W)$ for the Markov constraint $T \leftrightarrow Y \leftrightarrow W$ (see, e.g., Ref [39]).

## 2 Gibbs-posterior least squares regression

We consider one of the best-known learning algorithms: least squares linear regression. Not only is this algorithm widely used in practice, it has also proved a particularly well-suited setting for analyzing learning in the overparametrized regime [40–47]. Indeed it exhibits some of the most intriguing features of overparametrized learning, including benign overfitting and double descent

---

[2]The arguments of the logarithms in Eqs (6-7) are always nonnegative since the data processing inequality means that the IB problem is well-defined only for $\gamma > 1$, and the eigenvalues of a normalized regression matrix always range from zero to one [36].

[3]Note that the eigenvalues of $XX^\mathsf{T}$ and $X^\mathsf{T}X$ are identical except for the number of zero modes.

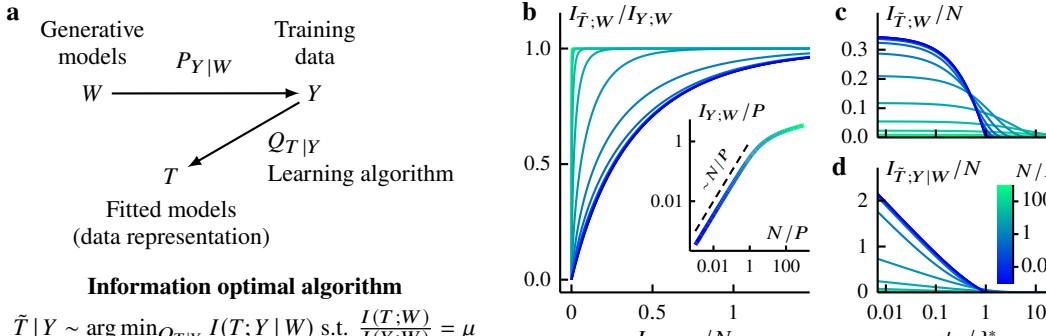

**Figure 1: Information optimal algorithm—a** A learning algorithm $Q_{T|Y}$ is a mapping from training data $Y$ to fitted models $T$. Information optimal algorithms minimize residual bits $I(T; Y \mid W)$—which are uninformative of the unknown generative model $W$—at fixed relevance level $\mu$, defined as the ratio between the encoded and available relevant bits, $I(T; W)$ and $I(Y; W)$. **b-d** The information content of optimal algorithms for learning a linear map (Sec 1.1) at various measurement densities $N/P$ (see color bar). **b** Optimal algorithms cannot increase the relevance level without encoding more residual bits. Increasing $N/P$ reduces the residual bits per sample but only when $N \lesssim P$. This results from the change in sample size dependence of relevant bits in the data from linear to logarithmic around $N \approx P$ (inset). That is, available relevant bits in each sample become redundant around $N \approx P$ and increasingly so as $N$ increases further. Learning algorithms use this redundancy to better distinguish signals from noise, thereby requiring fewer residual bits per sample. **c-d** The IB frontiers in (b) are parametrized by a spectral cutoff $\psi_c$ [see Eqs (8-9)]. Here we set $\omega^2/\sigma^2 = 1$ and let $P, N \to \infty$ at the same rate such that the ratio $N/P$ remains fixed and finite. The empirical spectral distribution $F^\Psi$ follows the standard Marchenko-Pastur law (see Sec 4).

which describe the surprisingly good generalization performance of overparametrized models and its nonmonotonic dependence on model complexity and sample size [31, 42, 48].

Inferring a model from data generally requires an assumption on a class of models, which defines the hypothesis space, as well as a learning algorithm, which outputs a hypothesis according to some criteria that rank how well each hypothesis explains the data. Linear regression restricts the model class to a linear map, parametrized by $T \in \mathbb{R}^P$, between an input $x_i$ and a predicted response $\hat{y}_i$,

$$\hat{y}_i = T \cdot x_i. \tag{12}$$

Minimizing the mean squared error $\frac{1}{N} \sum_{i=1}^{N} (\hat{y}_i - y_i)^2$ yields a closed form solution for the estimated regressor, $T^* = (XX^\mathsf{T})^{-1}XY$. However, this requires $XX^\mathsf{T}$ to be invertible and thus does not work in the overparametrized regime in which the sample covariance is not full rank and infinitely many models have vanishing mean squared error.

There are several approaches to break this degeneracy but perhaps the simplest and most studied is the ridge regularization which adds to the mean squared error the preference for model parameters with small $L_2$ norm, resulting in the regularized loss function

$$L(T, X, Y) = \frac{1}{N} \|Y - X^\mathsf{T}T\|_2^2 + \lambda \|T\|_2^2, \tag{13}$$

where $\lambda > 0$ controls the regularization strength. Minimizing this loss function leads to a unique solution $T_\lambda^* = (XX^\mathsf{T} + \lambda N I_P)^{-1}XY$ even when $N < P$.

**Gibbs posterior**—While ridge regression works in the overparametrized regime, it is a deterministic algorithm which does not readily lend itself to information-theoretic analyses because the mutual information between two deterministically related continuous random variables diverges. Instead we consider the Gibbs posterior (or Gibbs algorithm) which becomes a Gaussian channel when defined with the ridge regularized loss in Eq (13),

$$Q_{T|X,Y} \propto e^{-\beta L(T,Y,X)} \;\rightsquigarrow\; T \mid X, Y \sim N\left(\frac{1}{N}\frac{1}{\Psi + \lambda I_P}XY \,,\, \frac{1}{2\beta}\frac{1}{\Psi + \lambda I_P}\right). \tag{14}$$

Here $\beta$ denotes the inverse temperature. In the zero temperature limit $\beta \to \infty$, this algorithm returns the usual ridge regression estimate $T_\lambda^*$ (the mean of the above normal distribution) with probability

approaching one. Whilst randomized ridge regression needs not take the form above, Gibbs posteriors are attractive, not least because they naturally emerge, for example, from information-regularized risk minimization [5] (see also Ref [27] for a recent discussion).

**Markov constraint**—The generative model $P_{Y|W}$, true parameter distribution $P_W$ and learning algorithm $Q_{T|Y}$ [Eqs (3-4) & (14)] completely describe the relationship between all random variables of interest through the Markov factorization of their joint distribution (Fig 1a),

$$P_{T,Y,W} = P_W \otimes P_{Y|W} \otimes Q_{T|Y}. \tag{15}$$

Note that $P_{Y|W} = P_{Y|X,W}$ and $Q_{T|Y} = Q_{T|X,Y}$ in the fixed design setting (see Sec 1.1).

## 3  Information content of Gibbs regression

We now turn to the relevant and residual informations of the models that result from the Gibbs regression algorithm [Eq (14)]. Since all distributions appearing on the *rhs* of Eq (15) are Gaussian, the relevant and residual informations are given by

$$I(T; W) = \tfrac{1}{2} \ln \det \Sigma_T \Sigma_{T|W}^{-1} \quad \text{and} \quad I(T; Y \mid W) = \tfrac{1}{2} \ln \det \Sigma_{T|W} \Sigma_{T|Y}^{-1}. \tag{16}$$

Here we use the fact that $\Sigma_{T|W,Y} = \Sigma_{T|Y}$ due to the Markov constraint [Eq (15)]. The covariance $\Sigma_{T|Y}$ is defined by the learning algorithm in Eq (14). To obtain $\Sigma_{T|W}$ and $\Sigma_T$, we marginalize out $Y$ and $W$ in order from $P_{T,Y,W}$ [Eqs (3-4) & (14-15)] and obtain

$$T \mid W \sim N\left( \frac{\Psi}{\Psi + \lambda I_P} W, \ \frac{1}{2\beta} \frac{1}{\Psi + \lambda I_P} + \frac{\sigma^2}{N} \frac{\Psi}{(\Psi + \lambda I_P)^2} \right) \tag{17}$$

$$T \sim N\left( 0, \ \frac{1}{2\beta} \frac{1}{\Psi + \lambda I_P} + \frac{\sigma^2}{N} \frac{\Psi}{(\Psi + \lambda I_P)^2} + \frac{\omega^2}{P} \frac{\Psi^2}{(\Psi + \lambda I_P)^2} \right). \tag{18}$$

Substituting the covariance matrices above into Eq (16) yields (see Appendix B for derivation)

$$I(T; W) = \frac{P}{2} \int_{\psi > 0} dF^{\Psi}(\psi) \ \ln\left( 1 + \frac{\psi^2 / \lambda^*}{\psi + \frac{N}{2\beta\sigma^2}(\psi + \lambda)} \right) \tag{19}$$

$$I(T; Y \mid W) = \frac{P}{2} \int_{\psi > 0} dF^{\Psi}(\psi) \ \ln\left( 1 + \frac{2\beta\sigma^2}{N} \frac{\psi}{\psi + \lambda} \right). \tag{20}$$

The integration domains are restricted to positive real numbers since the eigenvalues of a covariance matrix are non-negative and the integrands vanish at $\psi = 0$.

In the zero temperature limit $\beta \to \infty$, the residual information diverges (as expected from a deterministic algorithm [23, 24]; see also Fig 2c) whereas the relevant information approaches the mutual information between the data $Y$ and the true parameter $W$,

$$I(T; W) \ \overset{\beta \to \infty}{\to} \ I(Y; W) = \frac{P}{2} \int_{\psi > 0} dF^{\Psi}(\psi) \ \ln(1 + \psi / \lambda^*). \tag{21}$$

Relevant and residual informations decrease with $\beta$ until they vanish as $\beta \to 0^+$ at which Gibbs posteriors become completely random (Fig 2a-c).

### 3.1  Zero temperature limit

At first sight it appears that our analyses are not applicable in the high-information limit since the residual information diverges for both the optimal algorithm and Gibbs regression (see Figs 1d & 2c). However the rates of divergence differ. Here we use this difference to characterize the efficiency of Gibbs regression in the zero temperature limit $\beta \to \infty$.

We first analyze the limiting behaviors of relevant information. From Eq (10), we see that the relevant information ratio of the IB solution approaches one at $\psi_c = 0$. Perturbing $\psi_c$ in Eq (8) away from zero results in a linear correction to the relevant information,

$$I(\tilde{T}; W) = I(Y; W) - \frac{\psi_c}{\lambda^*} \frac{P}{2} \int_{\psi > 0} dF^{\Psi}(\psi) + O(\psi_c^2). \tag{22}$$

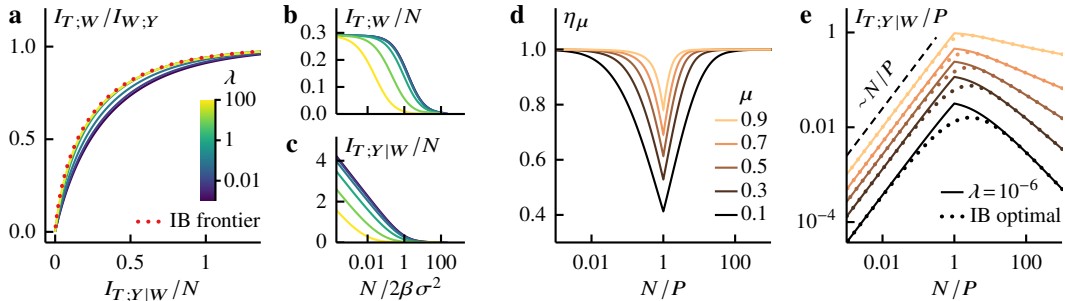

Figure 2: **Gibbs regression**—**a-c** The information content of Gibbs regression [Eq (14)] at $N/P = 1$ and various regularization strengths $\lambda$ (see color bar). **a** The information curves of Gibbs regression are bounded by the IB frontier (dotted curve). **b-c** The inverse temperature $\beta$ controls the stochasticity of Gibbs posteriors [Eqs (19-20)]. Both relevant and residual bits decrease with temperature and vanish in the limit $\beta \to 0$ where Gibbs posteriors become completely random. **d-e** Information efficiency [Eq (11)] and residual information of Gibbs regression with $\lambda = 10^{-6}$ vs measurement ratio at various relevance levels $\mu$ (see legend). **d** Gibbs regression approaches optimality in the limits, $N \gg P$ and $N \ll P$, and becomes least efficient at $N/P = 1$. **e** Residual bits of Gibbs regression and optimal algorithm (dotted) grow linearly with $N$ when $N \lesssim P$. This growth is similar to that of the available relevant bits (Fig 1b inset). But while the available relevant bits always increase with $N$, the residual bits *decrease* as $N$ exceeds $P$. Here we set $\omega^2/\sigma^2 = 1$ and let $P, N \to \infty$ at the same rate such that the ratio $N/P$ remains fixed and finite. The eigenvalues of the sample covariance follow the standard Marchenko-Pastur law (see Sec 4).

Keeping only the leading correction and recalling that $\mu = I(\tilde{T}; W)/I(Y; W)$ [Eq (11)], we obtain

$$\lim_{\mu \to 1} \frac{\psi_c}{\lambda^*} = \frac{\int_{\psi>0} dF^{\Psi}(\psi) \ln(1 + \psi/\lambda^*)}{\int_{\psi>0} dF^{\Psi}(\psi)} (1 - \mu). \tag{23}$$

Similarly expanding the relevant information of Gibbs regression [Eq (19)] around $\beta \to \infty$ yields

$$I(T; W) = I(Y; W) - \frac{N}{2\beta\sigma^2} \frac{P}{2} \int_{\psi>0} dF^{\Psi}(\psi) \frac{\psi + \lambda}{\psi + \lambda^*} + O(\beta^{-2}). \tag{24}$$

As a result, the correspondence between the low-temperature and high-information limits reads

$$\lim_{\mu \to 1} \frac{N}{2\beta\sigma^2} = \frac{\int_{\psi>0} dF^{\Psi}(\psi) \ln(1 + \psi/\lambda^*)}{\int_{\psi>0} dF^{\Psi}(\psi) \frac{\psi+\lambda}{\psi+\lambda^*}} (1 - \mu). \tag{25}$$

We turn to the residual bits. Expanding Eq (9) around $\psi_c = 0$ and Eq (20) around $\beta^{-1} = 0$ leads to

$$I(\tilde{T}; Y \mid W) = -\frac{P}{2} \int_{\psi>0} dF^{\Psi}(\psi) \ln \frac{\psi_c}{\lambda^*} + \frac{P}{2} \int_{\psi>0} dF^{\Psi}(\psi) \ln \frac{\psi}{\psi + \lambda^*} + O(\psi_c) \tag{26}$$

$$I(T; Y \mid W) = -\frac{P}{2} \int_{\psi>0} dF^{\Psi}(\psi) \ln \frac{N}{2\beta\sigma^2} + \frac{P}{2} \int_{\psi>0} dF^{\Psi}(\psi) \ln \frac{\psi}{\psi + \lambda} + O(\beta^{-1}) \tag{27}$$

From Eqs (23) & (25), we see that the residual informations above have the same logarithmic singularity, $\ln(1 - \mu)$, at $\mu = 1$. Therefore their difference remains finite even as $\mu \to 1$. Combining Eqs (23) & (25-27) and recalling our definition of information efficiency $\eta_\mu = I(\tilde{T}; Y \mid W)/I(T; Y \mid W)$ [Eq (11)] gives

$$\lim_{\mu \to 1} \eta_\mu = 1 - \frac{-1}{\ln(1 - \mu)} \left( \ln \frac{\int_{\psi>0} dF^{\Psi}(\psi) \frac{\psi+\lambda}{\psi+\lambda^*}}{\int_{\psi>0} dF^{\Psi}(\psi)} - \frac{\int_{\psi>0} dF^{\Psi}(\psi) \ln \frac{\psi+\lambda}{\psi+\lambda^*}}{\int_{\psi>0} dF^{\Psi}(\psi)} \right). \tag{28}$$

Note that Jensen's inequality guarantees that the terms in the parentheses sum to a non-negative value.

It is worth pointing out that, at $\lambda = \lambda^*$, the correction term in Eq (28) vanishes and the efficiency of deterministic Gibbs regression becomes minimally sensitive to algorithmic noise. Incidentally, this value of $\lambda$ also minimizes the $L_2$ prediction error of ridge regression in the asymptotic limit [40].

# 4 High dimensional limit

To place our results in the context of high dimensional learning, we specialize to the thermodynamic limit in which sample size and input dimension tend to infinity at a fixed ratio—that is, $N, P \to \infty$ at $N/P = n \in (0, \infty)$. While it is easy to grow the dimension of the true parameter $W$ (Sec 1.1), we have so far not specified how the design matrix $X$, and thus the training data $Y$, should scale in this limit.

To this end, we consider a setting in which the design matrix is generated from $X = \Sigma^{1/2} Z$ where $Z \in \mathbb{R}^{P \times N}$ is a matrix with iid entries drawn from a distribution with zero mean and unit variance, and $\Sigma \in \mathbb{R}^{P \times P}$ is a covariance matrix.[4] If $\Sigma$ admits a limiting spectral density as $P \to \infty$, then the empirical spectral distribution $F^\Psi$ becomes deterministic [49, 50].

To aid interpretation of our results, we frame all of the following discussions from the perspective that the input dimension $P$ is held fixed and a change in measurement density $n = N/P$ results only from a change in sample size $N$.

## 4.1 Isotropic covariates

For $\Sigma = I_P$, the empirical spectral distribution converges to to the standard Marchenko-Pastur law [49]

$$dF^\Psi(\psi) = n \frac{\sqrt{(\psi_+ - \psi)(\psi - \psi_-)}}{2\pi\psi} d\psi \quad \text{for} \quad \psi_- < \psi < \psi_+, \tag{29}$$

where $\psi_\pm = (1 \pm 1/\sqrt{n})^2$ and $F^\Psi(0) = \max(0, 1 - n)$. We use this spectral distribution in Figs 1-2.

*Optimal algorithm*—In Fig 1b, the IB optimal frontiers illustrate the fundamental trade-off; optimal algorithms cannot encode fewer residual bits without becoming less relevant. Figure 1c-d shows that encoded relevant and residual bits go down as $\psi_c$ increases and fewer eigenmodes contribute to the IB optimal representation [Eqs (8-9)]. However relevant and residual informations exhibit different behaviors at high information; as $\psi_c \to 0$, relevant information plateaus whereas residual information diverges logarithmically [see also Eq (26)].

*Gibbs regression*—Figure 2a depicts the information content of Gibbs regression at different regularization strengths [Eqs (19-20)] and illustrates the fundamental trade-off, similarly to the IB frontier (dotted) but at a lower relevance level. Here the information curves are parametrized by the inverse temperature $\beta$ which controls the algorithmic stochasticity; Gibbs posteriors become deterministic as $\beta \to \infty$ and completely random at $\beta = 0$ [Eq (14)]. In Fig 2b-c, we see that Gibbs regression encodes fewer relevant and residual bits as temperature goes up. Decreasing Gibbs temperature results in an increase in encoded information. In the zero-temperature limit, the relevant bits saturate while the residual bits grow logarithmically (cf. Fig 1c-d; see Sec 3.1 for a detailed analysis of this limit). The amount of encoded information depends also on the regularization strength $\lambda$. Figure 2b-c shows that, at a fixed temperature, an increase in $\lambda$ leads to less information extracted. However this does not necessarily mean that a larger $\lambda$ hurts information efficiency. Indeed a lower temperature can compensate for the decrease in information. In Fig 2a, we see that the information curves can be closer to the optimal frontier as $\lambda$ increases. In general the maximum efficiency occurs at an intermediate regularization strength that depends on data structure and measurement density (see also Sec 4.2).

*Efficiency*—Figure 2d displays the information efficiency of Gibbs regression at different relevant information levels (see Sec 1.3). We see that the efficiency approaches optimality ($\eta_\mu = 1$) in the limits $n \to 0$ and $n \to \infty$. Away from these limits, Gibbs regression requires more residual bits than the optimal algorithm to achieve the same level of relevance with an efficiency minimum around $n = 1$. We also see that the efficiency of Gibbs regression decreases with relevance level (see also Supplementary Figure in Appendix D).

*Extensivity*—Learning is qualitatively different in the over- and underparametrized regimes. In Fig 2e we see that both optimal algorithms and Gibbs regression exhibit nonmonotonic dependence on sample size. In the overparametrized regime $n < 1$, the residual information is *extensive* in sample size, i.e., it grows linearly with $N$. This scaling behavior mirrors that of the relevant bits in the data (Fig 1b inset). But unlike the available relevant bits which continue to grow in the data-abundant regime, albeit sublinearly—the encoded residual bits *decrease* with sample size in this limit (see also

---

[4]This prescription includes the case where input vectors are drawn iid from $x_i \sim N(0, \Sigma)$ for $i \in \{1, \ldots, N\}$.

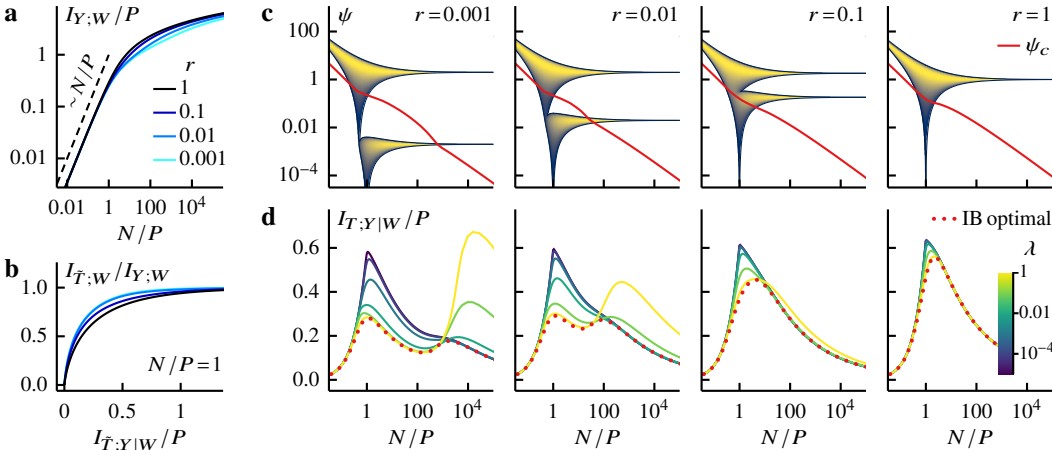

Figure 3: **Multiple descent under anisotropic covariates**—**a** The relevant bits in the data decreases slightly as the anisotropy ratio $r$ departs from one (see legend). When $n \lesssim 0$, the available relevant information grow linearly with $N$. Strong anisotropy sees this growth start becoming sublinear at smaller $N/P$. **b** The IB frontiers at $N/P = 1$. We see that while less relevant information is available in the anisotropic case, it takes fewer residual bits to achieve the same relevance level as the isotropic case (see legend in a). **c** The empirical spectral density of the sample covariance at different anisotropy ratios (see labels). Each vertical line is normalized by its maximum. We see that anisotropy splits the spectral continuum into two bands which merge into one as $N/P$ decreases. The solid line depicts the IB cutoff $\psi_c$ [Eqs (8-9)] for the relevance level $\mu = 0.8$. **e** The residual information of optimal algorithms (dotted) and Gibbs regression at various regularization strengths (see color bar) for $\mu = 0.8$ and different anisotropy ratios (same labels as in c). Here we set $\omega^2 / \sigma^2 = 1$ and let $P, N \to \infty$ at the same rate such that the ratio $N/P$ remains fixed and finite. The eigenvalues of the sample covariance follows the general Marchenko-Pastur theorem (see Sec 4.2).

Supplementary Figure in Appendix D). The resulting maximum is an information-theoretic analog of double descent—the decrease in overfitting level (test error) as the number of parameters exceeds sample size (decreasing $n$) [31, 48].

*Redundancy*—Indeed we could have anticipated the extensive behavior of the residual bits in the overparametrized regime (Fig 2e). In this limit, the extensivity of available relevant bits implies that the data encode relevant information with no redundancy. In other words, the relevant bits in one observation do not overlap with that in another. As a result, the dominant learning strategy is to treat each sample separately and extract the same amount of information from each of them, thus resulting in extensive residual information. In the data-abundant regime, on the other hand, the coding of relevant bits in the data becomes increasingly redundant (Fig 1b inset). Learning algorithms exploit this redundancy to better distinguish signals from noise, thereby encoding fewer residual bits.

## 4.2 Anisotropic covariates

To explore the effects of anisotropy, we consider a two-scale model in which the population spectral distribution $F^\Sigma$ is an equal mixture of two point masses at $s_+$ and $s_-$. We normalize the trace of the population covariance such that the signal variance, and thus the signal-to-noise ratio, does not depend on $F^\Sigma$—i.e., we set $\mathrm{tr}\, \Sigma / P = (s_+ + s_-)/2 = 1$ such that $\mathbb{E}[(W \cdot x_i)^2] = \mathbb{E}\|W\|^2 = \omega^2$. As a result the anisotropy in our two-scale model is parametrized completely by the eigenvalue ratio $r \equiv s_- / s_+$.

Unlike the isotropic case, the limiting empirical spectral distribution does not admit a closed form expression. We obtain $F^\Psi$ by solving the Silverstein equation and inverting the resulting Stieltjes transform [51] (see Appendix C). Figure 3c depicts the spectral density at various anisotropy ratios and measurement densities. At high measurement densities $n \gtrsim 1$, anisotropy splits the continuum part of the spectrum into two bands, corresponding to the two modes of the population covariance. These bands broaden as $n$ decreases and eventually merge into one in the overparametrized limit.

*Available information*—In Fig 3a, we see that anisotropy decreases the relevant information in the data, but does not affect its qualitative behaviors: the available relevant bits are extensive in the overparametrized regime and subextensive in the data-abundant regime. Although fewer relevant bits are available, learning needs not be less information efficient. Indeed the IB frontiers in Fig 3b illustrate that it takes fewer residual bits in the anisotropic case to reach the same level of relevance as in the isotropic case. This behavior is also apparent in Fig 3d (dotted) as we increase anisotropy levels (from right to left panels).

*Anisotropy effects*—Anisotropy affects optimal algorithms via the different scales in the population covariance. The signals along high-variance (easy) directions are stronger and, as a result, the coding of relevant bits in these directions becomes subextensive and redundant around $n \approx 1/2$ (the proportion of easy directions) instead of at one (Fig 3a). This earlier onset of redundancy allows for more effective signal-noise discrimination in the anisotropic case (Fig 3b & d). In the data-abundant limit, however, low-variance (hard) directions become important as learning algorithms already encode most of the relevant bits along the easy directions. Indeed the hard directions are harder for more anisotropic inputs and thus the required residual bits increase with anisotropy in the limit $n \to \infty$ (Fig 3d).

*Triple descent*—Perhaps the most striking effect of anisotropy is the emergence of an information-theoretic analog of (sample-wise) multiple descent, which describes disjoint regions where more data makes overfitting worse (larger test error) [32, 52]. In Fig 3d, we see that an additional residual information maximum emerges at large $n$. This behavior is a consequence of the separation of scales. The first maximum at $n \sim 1$ originates from easy directions and the other maximum at higher $n$ from hard directions. In fact the IB cutoff $\psi_c$ in Fig 3c demonstrates that the residual information maxima roughly coincide with the inclusion of all high-variance modes around $n \sim 1$ and low-variance modes at higher $n$.[5] In addition we note that for optimal algorithms the first maximum shifts to a lower $n$ as the anisotropy level increases. This observation is consistent with the fact that the onset of redundancy of relevant bits in the data occurs at smaller $n$ in the anisotropic case (Fig 3a).

*Gibbs regression*—Anisotropy makes Gibbs regression depend more strongly on regularization strengths, see Fig 3. In particular the information efficiency decreases with $\lambda$ near the first residual information minimum around $n \sim 1$ but this dependence reverses near the second maximum and at larger $n$. This behavior is expected. Inductive bias from strong regularization helps prevent noise from poisoning the models at small $n$. But when the data become abundant, regularization is unnecessary.

## 5   Conclusion & Outlook

We use the information bottleneck theory to analyze linear regression problems and illustrate the fundamental trade-off between relevant bits, which are informative of the unknown generative processes, and residual bits, which measure overfitting. We derive the information content of optimal algorithms and Gibbs posterior regression, thus enabling a quantitative investigation of information efficiency. In addition our analytical results on the zero temperature limit of the Gibbs posterior offer a glimpse of a connection between information efficiency and optimally tuned ridge regression. Finally, using results from random matrix theory, we reveal the information complexity of learning a linear map in high dimensions and unveil an information-theoretic analog of multiple descent phenomena. Since residual information is an upper bound on the generalization gap [4, 5], we believe that this information nonmonotonicity could be connected to the original double descent phenomena. But it remains to be seen how deep this connection is.

Our work paves the way for a number of different avenues for future research. While we only focus on isotropic regularization here, it would be interesting to understand how structured regularization affects information extraction. Information-efficiency analyses of different algorithms, such as Bayesian regression, and other classes of learning problems, e.g., classification and density estimation, are also in order. An investigation of information efficiency based on other $f$-divergence could lead to new insights into generalization. In particular an exact relationship exists between residual Jeffreys information and generalization error of Gibbs posteriors [27]. Finally exploring how coding redundancy in training data quantitatively affects learning phenomena in general would make for an exciting research direction (see, e.g., Refs [17, 19]).

---

[5]Note that Fig 3c does not show the zero modes which are present at $n < 1$. The fact that the spectral continuum appears to be above the IB cutoff at small $n$ does not mean all eigenmodes are used in the IB solution.

## Acknowledgments and Disclosure of Funding

This work was supported in part by the National Institutes of Health BRAIN initiative (R01EB026943), the National Science Foundation, through the Center for the Physics of Biological Function (PHY-1734030), the Simons Foundation and the Sloan Foundation.

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
