# Supplementary Material:
# Information bottleneck theory of high-dimensional regression: relevancy, efficiency and optimality

**Vudtiwat Ngampruetikorn,**[*] **David J. Schwab**
Initiative for the Theoretical Sciences, The Graduate Center, CUNY
[*]`vngampruetikorn@gc.cuny.edu`

## A   Information content of maximally efficient algorithms

Consider an IB problem where we are interested in an information efficient representation of $Y$ that is predictive of $W$ (Fig 1a). When $Y$ and $W$ are Gaussian correlated, the central object in constructing an IB solution is the normalized regression matrix $\Sigma_{Y|W}\Sigma_Y^{-1}$; in particular, its eigenvalues $\nu_i[\Sigma_{Y|W}\Sigma_Y^{-1}]$ completely characterize the information content of the IB optimal representation $\tilde{T}$ via (see Ref [1] for a derivation)

$$I(\tilde{T};W) = \frac{1}{2}\sum_{i=1}^{N} \max\left(0,\ \ln\frac{1-\gamma^{-1}}{\nu_i[\Sigma_{Y|W}\Sigma_Y^{-1}]}\right) \tag{1}$$

$$I(\tilde{T};Y \mid W) = \frac{1}{2}\sum_{i=1}^{N} \max(0,\ \ln(\gamma(1-\nu_i[\Sigma_{Y|W}\Sigma_Y^{-1}]))), \tag{2}$$

where $N$ is the dimension of $Y$ and $\gamma$ parametrizes the IB trade-off [Eq (1)].

Our work focuses on the following generative model for $W$ and $Y$ (see Sec 1.1)

$$W \sim N(0, \tfrac{\omega^2}{P}I_P) \quad \text{and} \quad Y \mid W \sim N(X^\mathsf{T}W, \sigma^2 I_N). \tag{3}$$

Marginalizing out $W$ yields

$$Y \sim N(0, \sigma^2 I_N + \tfrac{1}{P}X^\mathsf{T}X). \tag{4}$$

As a result, the normalized regression matrix reads

$$\Sigma_{Y|W}\Sigma_Y^{-1} = \sigma^2 I_N \frac{1}{\sigma^2 I_N + \frac{1}{P}X^\mathsf{T}X} = \left(I_N + \frac{1}{\lambda^*}\frac{X^\mathsf{T}X}{N}\right)^{-1} \quad \text{where} \quad \lambda^* \equiv \frac{P}{N}\frac{\sigma^2}{\omega^2}. \tag{5}$$

Substituting Eq (5) into Eqs (1-2) gives

$$I(\tilde{T};W) = \frac{1}{2}\sum_{i=1}^{N} \max\left(0,\ \ln\left((1-\gamma^{-1})(1+\phi_i[X^\mathsf{T}X/N]/\lambda^*)\right)\right) \tag{6}$$

$$I(\tilde{T};Y \mid W) = \frac{1}{2}\sum_{i=1}^{N} \max\left(0,\ \ln\frac{\gamma\phi_i[X^\mathsf{T}X/N]}{\lambda^* + \phi_i[X^\mathsf{T}X/N]}\right), \tag{7}$$

where $\phi_i[X^\mathsf{T}X/N]$ denote the eigenvalues of $X^\mathsf{T}X/N$. Since the eigenvalues of $X^\mathsf{T}X/N$ and the sample covariance $\Psi = XX^\mathsf{T}/N$ are identical except for the zero modes which do not contribute to information, we can recast the above equations as

$$I(\tilde{T};W) = \frac{1}{2}\sum_{i=1}^{P} \max\left(0,\ \ln(1-\gamma^{-1})(1+\psi_i/\lambda^*)\right) \tag{8}$$

$$I(\tilde{T};Y \mid W) = \frac{1}{2}\sum_{i=1}^{P} \max\left(0,\ \ln\frac{\gamma\psi_i}{\lambda^* + \psi_i}\right), \tag{9}$$

36th Conference on Neural Information Processing Systems (NeurIPS 2022).

where $\psi_i$ are the eigenvalues of $\Psi$ and the summation limits change to $P$, the number of eigenvalues of $\Psi$. Introducing the cumulative spectral distribution $F^\Psi$ and replacing the summations with integrals results in

$$I(\tilde{T}; W) = \frac{P}{2} \int dF^\Psi(\psi) \max\left(0, \ln\left((1 - \gamma^{-1})(1 + \psi/\lambda^*)\right)\right) \tag{10}$$

$$I(\tilde{T}; Y \mid W) = \frac{P}{2} \int dF^\Psi(\psi) \max\left(0, \ln \frac{\gamma\psi}{\lambda^* + \psi}\right). \tag{11}$$

We see that the contributions to the integrals come from the logarithms but only when they are positive. This condition can be recast into integration limits (note that $\gamma > 0$ and $\lambda^* > 0$)

$$\ln\left((1 - \gamma^{-1})(1 + \psi/\lambda^*)\right) > 0 \implies \psi > \lambda^*/(\gamma - 1) \tag{12}$$

$$\ln \frac{\gamma\psi}{\lambda^* + \psi} > 0 \implies \psi > \lambda^*/(\gamma - 1). \tag{13}$$

Finally we define the lower cutoff $\psi_c \equiv \lambda^*/(\gamma - 1)$ and use the above limits to rewrite the expressions for relevant and residual informations,

$$I(\tilde{T}; W) = \frac{P}{2} \int_{\psi > \psi_c} dF^\Psi(\psi) \ln \frac{\psi + \lambda^*}{\psi_c + \lambda^*} = \frac{P}{2} \int_{\psi > \psi_c} dF^\Psi(\psi) \ln\left(1 + \frac{\psi - \psi_c}{\psi_c + \lambda^*}\right) \tag{14}$$

$$I(\tilde{T}; Y \mid W) = \frac{P}{2} \int_{\psi > \psi_c} dF^\Psi(\psi) \ln \frac{\psi}{\psi_c} \frac{\psi_c + \lambda^*}{\psi + \lambda^*} = \frac{P}{2} \int_{\psi > \psi_c} dF^\Psi(\psi) \ln \frac{\psi}{\psi_c} - I(\tilde{T}; W). \tag{15}$$

These equations are identical to Eqs (8-9) in the main text.

## B  Information content of Gibbs-posterior regression

To compute the information content of Gibbs regression [Eq (14)], we first recall that the mutual information between two Gaussian correlated variables, $A$ and $B$, is given by

$$I(A; B) = \frac{1}{2} \ln \det \Sigma_A \Sigma_{A|B}^{-1}, \tag{16}$$

where $\Sigma_A$ is the covariance of $A$, and $\Sigma_{A|B}$ of $A \mid B$.

We now write down the relevant information, using the covariances $\Sigma_{T|W}$ and $\Sigma_T$ from Eqs (17-18),

$$I(T; W) = \frac{1}{2} \ln \det \left(\Sigma_T \Sigma_{T|W}^{-1}\right) \tag{17}$$

$$= \frac{1}{2} \ln \det \frac{\frac{1}{2\beta} \frac{1}{\Psi + \lambda I_P} + \frac{\sigma^2}{N} \frac{\Psi}{(\Psi + \lambda I_P)^2} + \frac{\omega^2}{P} \frac{\Psi^2}{(\Psi + \lambda I_P)^2}}{\frac{1}{2\beta} \frac{1}{\Psi + \lambda I_P} + \frac{\sigma^2}{N} \frac{\Psi}{(\Psi + \lambda I_P)^2}} \tag{18}$$

$$= \frac{1}{2} \ln \det \left(I_P + \frac{\Psi^2/\lambda^*}{\Psi + \frac{N}{2\beta\sigma^2}(\Psi + \lambda I_P)}\right) \tag{19}$$

$$= \frac{1}{2} \operatorname{tr} \ln \left(I_P + \frac{\Psi^2/\lambda^*}{\Psi + \frac{N}{2\beta\sigma^2}(\Psi + \lambda I_P)}\right) \tag{20}$$

$$= \frac{1}{2} \sum_{i=1}^{P} \ln \left(1 + \frac{\psi_i^2/\lambda^*}{\psi_i + \frac{N}{2\beta\sigma^2}(\psi_i + \lambda)}\right) \tag{21}$$

$$= \frac{P}{2} \int_{\psi > 0} dF^\Psi(\psi) \ln \left(1 + \frac{\psi^2/\lambda^*}{\psi + \frac{N}{2\beta\sigma^2}(\psi + \lambda)}\right), \tag{22}$$

where $\lambda^* = P\sigma^2/N\omega^2$. In the above, we use the identity $\ln \det H = \operatorname{tr} \ln H$ which holds for any positive-definite Hermitian matrix $H$, let $\psi_i$ denote the eigenvalues of the sample covariance $\Psi$ and introduce $F^\Psi$, the cumulative distribution of eigenvalues. We also assume that $\lambda$ and $\beta$ are finite

and positive. Note that the integral is limited to positive real numbers because the eigenvalues of a covariance matrix is non-negative and the integrand vanishes for $\psi = 0$.

Following the same logical steps as above and noting that the Markov constraint $W \leftrightarrow Y \leftrightarrow T$ implies $\Sigma_{T|Y,W} = \Sigma_{T|Y}$, we write down the residual information,

$$I(T;Y \mid W) = \frac{1}{2} \ln \det \left( \Sigma_{T|W} \Sigma_{T|Y,W}^{-1} \right) \tag{23}$$

$$= \frac{1}{2} \ln \det \left( \Sigma_{T|W} \Sigma_{T|Y}^{-1} \right) \tag{24}$$

$$= \frac{1}{2} \ln \det \left( \frac{\frac{1}{2\beta} \frac{1}{\Psi+\lambda I_P} + \frac{\sigma^2}{N} \frac{\Psi}{(\Psi+\lambda I_P)^2}}{\frac{1}{2\beta} \frac{1}{\Psi+\lambda I_P}} \right) \tag{25}$$

$$= \frac{P}{2} \int_{\psi>0} dF^\Psi(\psi) \ln \left( 1 + \frac{2\beta\sigma^2}{N} \frac{\psi}{\psi+\lambda} \right) \tag{26}$$

where we use the covariance matrices $\Sigma_{T|W}$ and $\Sigma_{T|Y}$ from Eqs (17) & (14).

## C  Marchenko-Pastur law

Consider $X = \Sigma^{1/2} Z$ where $Z \in \mathbb{R}^{P \times N}$ is a matrix with iid entries drawn from a distribution with zero mean and unit variance, and $\Sigma \in \mathbb{R}^{P \times P}$ is a covariance matrix. In addition we take the asymptotic limit $N \to \infty$, $N \to \infty$ and $P/N \to \alpha \in (0, \infty)$. If the population spectral distribution $F^\Sigma$ converges to a limiting distribution, the spectral distribution of the sample covariance $\Psi = XX^\mathsf{T}/N$ becomes deterministic [2]. The density, $f^\Psi(\psi) = dF^\Psi(\psi)/d\psi$, is related to its Stieltjes transform $m(z)$ via

$$f^\Psi(\psi) = \frac{1}{\pi} \operatorname{Im} m(\psi + i\,0^+), \quad \psi \in \mathbb{R}. \tag{27}$$

We can obtain $f^\Psi$ by solving the Silverstein equation for the companion Stieltjes transform $v(z)$ [3],

$$-\frac{1}{v(z)} = z - \alpha \int_{\mathbb{R}^+} dF^\Sigma(s) \frac{s}{1 + sv(z)}, \quad z \in \mathbb{C}^+, \tag{28}$$

and using the relation

$$m(z) = \alpha^{-1}(v(z) + z^{-1}) - z^{-1}. \tag{29}$$

Here $\mathbb{C}^+$ denotes the upper half of the complex plane.

# D  Supplementary figure

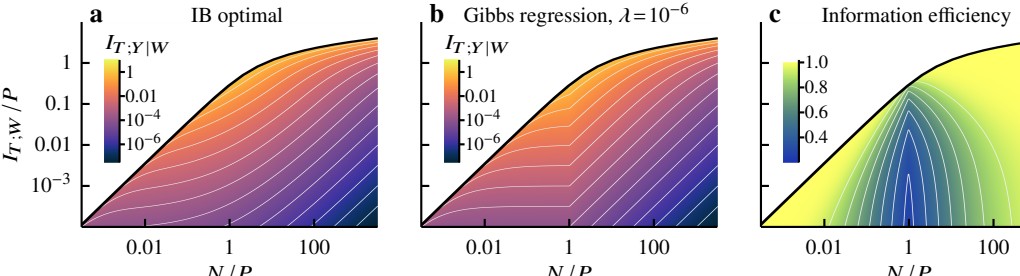

Figure 1: Gibbs ridge regression is least information efficient around $N/P = 1$. **a** Residual information $I(T;Y \mid W)$ of the IB optimal algorithm over a range of sample densities $N/P$ (horizontal axis) and given extracted relevant bits $I(T;W)$ (vertical axis). The extracted relevant bits are bounded by the available relevant bits in the data (black curve), i.e., the data processing inequality implies $I(T;W) \leq I(Y;W)$. **b** Same as (a) but for Gibbs regression with $\lambda = 10^{-6}$. Holding other things equal, Gibbs regression estimators encode more residual bits than optimal representations. **c** Information efficiency, the ratio between residual bits in optimal representations (a) and Gibbs estimator (b), is minimum around $N/P = 1$. Here we set $\omega^2/\sigma^2 = 1$ and let $P, N \to \infty$ at the same rate such that the ratio $N/P$ remains fixed and finite. The eigenvalues of the sample covariance follow the standard Marchenko-Pastur law (see Sec 4).