# OpenReview forum: "Information bottleneck theory of high-dimensional regression: relevancy, efficiency and optimality"
_NeurIPS.cc/2022/Conference — NeurIPS 2022 Accept_

### Official Review · Reviewer_Lvpt · 2022-07-09

**Rating:** 5
**Confidence:** 4
**Soundness:** 3 good
**Presentation:** 2 fair
**Contribution:** 2 fair

**Summary:**

The paper studies overfitting in linear regression using information bottleneck and Gibbs-posterior least squares regression. The authors define information efficiency in equation (11), which is a ratio of conditional MIs between the optimal solution of Gaussian variables given in reference [36] and an algorithm of interest. Since linear regression objective requires $XX^T$ to be invertible, and ridge regularization yields deterministic solution, the authors propose to use Gibbs posterior (Eq (14)). The information efficiency is derived in Eq (28). They further specialize it to the thermodynamic limit.

** After reading the author rebuttal, where additional information on the assumptions and conceptual framework are provided. I agree that this could be a positive first step towards understanding overfitting using IB. My main concern is about the information efficiency defined in Eq (11). As the authors have provided more detail on the motivation of this definition, I am willing to raise my score.**

**Questions:**

The assumptions require the design matrix to be fixed and that the error follows Gaussian distribution. Would this affect the overfitting?

It is hard to obtain information theoretic results for the linear regression and ridge regularization in Eq (12) and Eq (13). Therefore, the authors apply Gibbs posterior. How would the results deviate from the original linear regression model?

**Strengths And Weaknesses:**

Strengths:
IB has received a lot of attentions in the study of neural networks. Linear regression using Gaussian channels has also been extensively studied. This paper provides an idea of using Gibbs posterior to obtain the information theoretic analysis of overfitting in linear regression.

Weakness:
It is not clear to me why the information efficiency defined in Eq (11) is useful for the study of overfitting. In NNs, the IB objective is derived from a rate-distortion problem. Here, if the goal is to achieve good generalization, the objective should be derived from optimizing generalization errors. Moreover, some assumptions are restrictive and not well justified. See questions below for other comments.

---

> ### Author Response · Authors · 2022-08-02
> **Author Response (1/2)**
>
> Thank you for your valuable comments and questions! We take them seriously and respond to each of them point by point below. Please, let us know if we misunderstood any of your points.
>
> &ensp;
>
> >__Strengths And Weaknesses:__
>
> >_It is not clear to me why the information efficiency defined in Eq (11) is useful for the study of overfitting. In NNs, the IB objective is derived from a rate-distortion problem. Here, if the goal is to achieve good generalization, the objective should be derived from optimizing generalization errors._
>
> Overfitting of course describes scenarios in which fitted models depend too much on noise in the training data, resulting in suboptimal generalization, but how do we quantify this dependence? And how much is too much? Our work offers an attractive answer to these two questions.
>
> First, we use mutual information as a measure of statistical dependence between random variables of interest. In particular, the residual information, I(T;Y|W)=I(T;Y)-I(T;W), quantifies how much fitted models depend on noise (the bits that are not informative of the generative process). In addition, I(T;Y|W) is formally related to the generalization gap, see, e.g., Russo & Zou (2016) and Xu & Raginsky (2017). However, mutual information is more general than a particular loss function or performance metric.
>
> Second, we use the IB method to derive the minimum achievable residual bits; all learning algorithms must encode at least this much information about noise. This minimum value provides a baseline for how much fitted models _have to_ depend on noise. The information efficiency in Eq (11) offers an intuitive measure of how far a fitted model is from this baseline, and is normalized to be between 0 and 1 in line with standard definitions of efficiency.
>
> We note further that although generalization gaps somewhat measure how much models depend on noise, they are based on problem-specific performance metrics and it is unclear how we can systematically compute the minimum achievable generalization gaps, which in general may not be differentiable. In contrast, relevant and residual information and the IB principle are not problem specific, and therefore our conceptual framework offers a step towards a unifying understanding of overfitting.
>
> We emphasize that we do not propose a new machine-learning method based on IB. Our goal is to formulate a new setting in which we can interrogate learning problems and uncover and interpret new interesting phenomenology from a new perspective. In our view, understanding the phenomenology of high-dimensional learning from different perspectives is key to unlocking new insights. We believe our work offers a step in this direction.
>
> &ensp;
>
> >_Moreover, some assumptions are restrictive and not well justified._
>
> Since this comment is related to that of Reviewer L3Rw, we repeat the relevant responses below.
>
> Although linear models are simple, they capture many important features of high-dimensional learning, of which our understanding is lacking, and can serve as a useful (if simple) setting for theoretical investigations. In fact, even linear regression is far from being well-understood; it still generates new surprising results (Hastie et al., 2022) and provides much needed insights into high-dimensional learning, see, e.g., Bartlett et al. (2020), Wu & Xu (2020), Richards et al. (2021) and Mel & Ganguli (2021). In addition, linear models are instrumental in building our understanding of deep networks, see, e.g., Saxe et al. (2014), Lampinen & Ganguli (2019), Ji & Telgarsky (2019) and Arora et al. (2019).
>
> We would like to further emphasize that our main contribution is the formulation of a new and informative framework for investigating learning problems. In particular, we demonstrate how the concepts from IB can be used to analyze learning algorithms, and we identify and interpret new information-theoretic phenomenology. Linear regression is very well suited for our purposes. In fact, we think of the theoretical tractability of linear settings as a feature that allows us to introduce new concepts, apply them to practical algorithms and discuss the many facets of high-dimensional learning (including overparametrization, regularization, double descent and covariance anisotropy) through the lens of information theory. While we focus on linear regression in this work, our framework and the concepts we introduce apply to more complicated learning scenarios. One possible generalization would be to consider random feature models (Rahimi & Recht, 2007) for which available technical tools and results should make information-theoretic analyses tractable, see, e.g., Mei & Montanari (2019) and Mel & Pennington (2022) and references therein.
>
> &ensp;
>
> *continued in next comment*

---

> ### Author Response · Authors · 2022-08-02
> **Author Response (2/2)**
>
> _continued from previous comment_
>
> &ensp;
>
> >__Questions:__
>
> >_The assumptions require the design matrix to be fixed and that the error follows Gaussian distribution._
>
> We stress that the fixed-design setting is arguably the predominant view of linear regression [see, e.g., Chapter 3 in Hastie et al. (2009)]. In addition, random matrix universality means that our thermodynamic-limit results remain the same for any design matrix so long as it is generated from the process described in Sec 4 (line 175).
>
> We also note that Gaussian noise is a standard assumption, see, e.g., Mel & Ganguli (2021), Chen et al. (2021), Tripuraneni et al. (2021), and Koehler et al. (2021). It leads to a theoretically tractable setup that enables systematic exploration of high-dimensional learning (see also our response above).
>
> >_Would this affect the overfitting?_
>
> While the exact quantitative results (the values of relevant and residual information and information efficiency) depend on the specific details of the setup, the conceptual formulation and interpretation do not. Relaxing the fixed-design and Gaussian noise assumptions would make for an interesting study, albeit requiring more advanced tools from multivariate statistics and random matrix theory; however, in our opinion, these standard assumptions suit our primary goal which is to introduce an information-theoretic view of overfitting and provide a proof-of-principle analysis. Please, let us know if we misunderstood your question.
>
> >_It is hard to obtain information theoretic results for the linear regression and ridge regularization in Eq (12) and Eq (13). Therefore, the authors apply Gibbs posterior. How would the results deviate from the original linear regression model?_
>
> It is actually _impossible_ to directly perform the same information-theoretic analysis for deterministic (ridge) regression. This is because the mutual information between two deterministically related continuous random variables diverges (see line 123). What we can do is taking the noiseless limit of the Gibbs posterior, see Sec 3.1. In this limit, Gibbs regression yields the ridge regression estimate [from minimizing Eq (13)] with probability approaching one (line 126). In Sec 3.1, we derive relevant information, residual information, and information efficiency for this limiting case.
>
> &ensp;
>
> __References:__  (in order of appearance)
> - Russo & Zou, Controlling Bias in Adaptive Data Analysis Using Information Theory, AISTATS (2016)
> - Xu & Raginsky, Information-theoretic analysis of generalization capability of learning algorithms, NeurIPS (2017)
> - Hastie, Montanari, Rosset & Tibshirani, Surprises in high-dimensional ridgeless least squares interpolation, The Annals of Statistics 50, 949 (2022)
> - Bartlett, Long, Lugosi & Tsigler, Benign overfitting in linear regression, PNAS 117, 30063 (2020)
> - Wu & Xu, On the Optimal Weighted l2 Regularization in Overparameterized Linear Regression, NeurIPS (2020)
> - Richards, Mourtada & Rosasco, Asymptotics of Ridge(less) Regression under General Source Condition, AISTATS (2021)
> - Mel & Ganguli, A theory of high dimensional regression with arbitrary correlations between input features and target functions: sample complexity, multiple descent curves and a hierarchy of phase transitions, ICML (2021)
> - Saxe, McClelland & Ganguli, Exact solutions to the nonlinear dynamics of learning in deep linear neural networks, ICLR (2014)
> - Lampinen & Ganguli, An analytic theory of generalization dynamics and transfer learning in deep linear networks, ICLR (2019)
> - Ji & Telgarsky, Gradient descent aligns the layers of deep linear networks, ICLR (2019)
> - Arora, Cohen, Golowich & Hu, A Convergence Analysis of Gradient Descent for Deep Linear Neural Networks, ICLR (2019)
> - Rahimi & Recht, Random Features for Large-Scale Kernel Machines, NeurIPS (2007)
> - Mei & Montanari, The generalization error of random features regression: Precise asymptotics and double descent curve, arXiv:1908.05355
> - Mel & Pennington, Anisotropic Random Feature Regression in High Dimensions, ICLR (2022)
> - Hastie, Tibshirani & Friedman, The Elements of Statistical Learning, 2nd ed. (Springer, 2009)
> - Chen, Min, Belkin & Karbasi, Multiple Descent: Design Your Own Generalization Curve, NeurIPS (2021)
> - Tripuraneni, Adlam & Pennington, Overparameterization Improves Robustness to Covariate Shift in High Dimensions, NeurIPS (2021)
> - Koehler, Zhou, Sutherland & Srebro, Uniform Convergence of Interpolators: Gaussian Width, Norm Bounds and Benign Overfitting, NeurIPS (2021)

---

### Official Review · Reviewer_jTSy · 2022-07-11

**Rating:** 6
**Confidence:** 3
**Soundness:** 3 good
**Presentation:** 2 fair
**Contribution:** 4 excellent

**Summary:**

This paper relies on information theory to investigate linear regressions, analysing linear regressors through an information bottleneck lens.
Basically, if we have a fixed input matrix X and we imagine a process where we:
1. Sample a generative linear parameter vector W;
2. Sample a set of targets Y according to W*X + GaussianNoise;
3. Learn a vector T, which tries to reconstruct W from both X and Y.

We can decompose the amount of information T encodes about the training data (X; Y) as:
I(T; Y) = I(T; Y | W) + I(T; W)

Note that X does not appear in these equations, as it is assumed to be fixed.

The paper then gives interpretations to each of these mutual information terms and analyses them: I(T; W) is the relevant learned bits, and I(T; Y | W) is the residual bits.

Using these terms the paper first analyses the behavior of an information theoretically optimal algorithm, i.e. one which minimises the term I(T; Y | W) subject to I(T; W)/I(Y; W)=\mu for several different values of \mu.
They then present the trade-off of how much relevant vs residual information a model can extract from Y when given a dataset and parameters of size N/P.

The paper then analyses the information content of a learning algorithm which fits a linear regression using a Gibbs posterior.
They show Gibbs regression is optimal when N << P or N >> P, where optimality is measured in terms of I(T_{optimal}; Y | W) / I(T_{Gibbs}; Y | W)---i.e. how much more residual information the Gibbs procedure extracts than an optimal method---subject to it extracting the same information about the parameters I(T_{optimal}; W) = I(T_{Gibbs}; W).
The paper than shows that the amount of residual information per parameter a Gibbs regression extracts I(T; Y | W) / P first grows with N/P when N<P and then decreases for N>P. It then relates these results to the double descent phenomenon.



========= Update post author--reviewer discussion ============

As noted in my review and responses. The main issue with the paper (in my reading) is that presenting results which control for $\mu=\frac{I(T; W)}{I(Y; W)}$ being constant---as opposed to $I(T; W)$---could lead to statistical artifacts which superficially seem interesting, but which do not reflect deeper insights into how these models work. Further, the authors arguments in favour of continuing to control only for $\mu=\frac{I(T; W)}{I(Y; W)}$ did not convince me.

In the authors' last response, however, they pointed me to the results which I requested (i.e., the ones controlling for $I(T; W)$ being constant) and the results seem to hold there. I thus remove my objection against this paper being accepted and raise my score to a 6.

In case this paper gets accepted, I would suggest the authors create a plot showing the model's performance (e.g., expected mean squared error) as a function of both $I(T; W)$ and $I(T; Y | W)$. This could make clearer the connection between model performance and both residual and relevant information.

**Questions:**

* What does the arrow notation in equation 23/25/28 represent?

* Do you think the results presented in Figures 2d and 2e could be a byproduct of plotting those curves for fixed \mu values? Why should we interpret a simultaneous reduction in I(T; W) and  I(T; Y | W) for N<<P as being related to the double descent phenomenon?

* If I understood correctly, in all plots where the x-axis is N/P only the N changes, right? Why display it as N/P instead of N then?

* Figure captions say "Here we set [...] and let 𝑃, 𝑁 →∞." What does it mean to let P, N -> \infty, while only N/P changes in the x-axis?

* Nitpick: Equation 1 discusses random variables S, T, and W . Why use S instead of Y?

**Limitations:**

N/A.

**Strengths And Weaknesses:**

This paper's strengths are:
* This paper is very interesting and relevant in its chosen topic of study;
* Framing the information bottleneck in terms of the amount of information the learned parameters T hold about the generative parameters W (instead of the typical way of discussing how a set of representations Z hold information about the target Y vs the input X) seems new to me, although it was not presented as such.
* Relating the information bottleneck theory to double descent could be very impactful and an interesting way to interpret that phenomenon.

The paper's weaknesses are:
* Some of the results in the paper---Figures 2d, and 2e, i.e. the ones related to the double descent phenomenon---are (perhaps) unintuitive. I tried to analyse them carefully, but I am still not convinced about them. Specifically, while the authors interpret these plots as providing an information-theoretic explanation for double descent, I think the shape of these curves might simply be a byproduct of the experiment design used to plot them. Specifically, each of the colored curves are conditioned such that  \mu=I(T; W)/I(Y; W) is constant. The curves then show that, as models get to the regime of N<P, there is a reduction on the residual information I(T; Y | W). The paper then relates this to double descent. However, couldn't this be just a byproduct of a reduction in I(Y; W)? In other words, to keep \mu constant the curves are displaying as a single thing the results for models with very different "performances"/relevant information I(T; W)---might this be what is driving the curves to look like this?
* In my opinion, the paper's presentation in Section 3.1 is lacking. While throughout most of the paper the authors try to motivate and explain the equations they present, Section 3.1 is harder to follow and the equations there are less well described. Further, I did not fully understand what the arrow notation in equation 23/25/28 represent. Are those limits?

---

> ### Author Response · Authors · 2022-08-02
> **Author Response (1/2)**
>
> Thank you for highlighting the relevancy, novelty and potential impact of our work! We also appreciate the helpful comments which led us to improve the clarity and presentation of our paper.
>
> > __Strengths And Weaknesses:__
>
> >_[...] the authors interpret these plots as providing an information-theoretic explanation for double descent [...]_
>
> We actually do not claim that our results _explain_ double descent. What we point out in the manuscript is the existence of information-theoretic analogs of such phenomena. To the best of our knowledge, our work is the first to demonstrate that information-theoretic measures can exhibit nonmonotonic sample size dependence. Since residual information is an upper bound on the generalization gap (Russo & Zou, 2016; Xu & Raginsky, 2017), we believe that this information nonmonotonicity could be connected to the original double descent phenomena. But it remains to be seen how deep this connection is. We will clarify this and highlight it as a promising avenue for future research in our updated paper.
>
> &ensp; __References:__
> - Russo & Zou, Controlling Bias in Adaptive Data Analysis Using Information Theory, AISTATS (2016)
> - Xu & Raginsky, Information-theoretic analysis of generalization capability of learning algorithms, NeurIPS (2017)
>
> >_I think the shape of these curves might simply be a byproduct of the experiment design used to plot them. Specifically, each of the colored curves are conditioned such that \mu=I(T; W)/I(Y; W) is constant. The curves then show that, as models get to the regime of N<P, there is a reduction on the residual information I(T; Y | W). The paper then relates this to double descent. However, couldn't this be just a byproduct of a reduction in I(Y; W)? In other words, to keep \mu constant the curves are displaying as a single thing the results for models with very different "performances"/relevant information I(T; W)---might this be what is driving the curves to look like this?_
>
> While we agree that each curve in Fig 2d&e compares models with different relevant bits, this does not mean the comparison is unfair. Most deterministic algorithms extract all of the relevant information available in the data and would have a fixed $\mu = 1$, regardless of sample size. For example, if we take the deterministic limit of Gibbs regression (let $\beta\to\infty$), we see that I(T;W) = I(Y;W) and $\mu = 1$, see Eq (24). Since the available relevant bits I(Y;W) grows with N (Fig 1b inset), it is easier to achieve high I(T;W) at large N than at small N. In our view, a fair comparison should take this fact into account, and fixing the relative relevance is an intuitive option.
>
> To further clarify this point, we have included a new figure in the Supplementary Material (which we will add to the Appendix of the final version of the paper and refer to from the main text). This figure shows the residual bits I(T;Y|W)/P as a function of extracted relevant bits I(T;W)/P and measurement density N/P for IB optimal and Gibbs algorithms (Panels a & b, respectively). For Gibbs regression, we also depict the information efficiency as a function of I(T;W)/P and N/P (Panel c). We see that fixing I(T;W) (vertical axis) would limit the range of N/P since I(T;W) <= I(Y;W) (data processing inequality) and I(Y;W) depends on N/P (black curve). At fixed extracted relevant bits I(T;W), residual bits I(T;Y|W) decrease with N/P monotonically but at different rates depending on whether N/P < 1 or N/P > 1 (Panels a & b). In Panel c, we see that the information efficiency of Gibbs regression is lowest around N/P = 1 and is nonmonotonic in sample size, regardless of whether we fix I(T;W) or $\mu$.
>
> >_In my opinion, the paper's presentation in Section 3.1 is lacking. While throughout most of the paper the authors try to motivate and explain the equations they present, Section 3.1 is harder to follow and the equations there are less well described. Further, I did not fully understand what the arrow notation in equation 23/25/28 represent. Are those limits?_
>
> Thank you for pointing this out&mdash;we will absolutely improve the presentation of our work. The equations in Sec 3.1 are the limiting cases of residual and relevant information, derived in previous sections. We will expand and revise this section in the updated manuscript to give more intuition behind each equation. Please, see below regarding the arrows.
>
> >__Questions:__
>
> >_What does the arrow notation in equation 23/25/28 represent?_
>
> We apologize for this confusion. The arrows are indeed limits. We will make sure to clarify these equations in the revised manuscript.
>
> >_Do you think the results presented in Figures 2d and 2e could be a byproduct of plotting those curves for fixed \mu values? Why should we interpret a simultaneous reduction in I(T; W) and I(T; Y | W) for N<<P as being related to the double descent phenomenon?_
>
> Please see our responses above.
>
> &ensp;
>
> _continued in next comment_

---

> > ### Comment · Reviewer_jTSy · 2022-08-05
> > **Answer to Authors**
> >
> > I thank the authors for their response.
> >
> > While I have read their response, I was not convinced by their argument and I am thus keeping my score.
> >
> > Specifically, I am not convinced this is an analog of the double descent phenomenon, because of the reasons I pointed out in my review.
> > Namely, the figures show a "double descent" analog while holding $\mu=I(T; W)/I(Y; W)$ fixed.
> > For each point in these curves, though, I(Y; W) changes---and thus so does I(T; W).
> > Double descent, however, displays a decrease/increase in the model's performance (whose analogous would be I(T; W)) as the model reaches $N/P = 1$; and not of model relative performance vs a theoretical maximum (whose analogous would be I(T; W)/I(T; W)).

---

> > > ### Author Response · Authors · 2022-08-07
> > > **Author Response**
> > >
> > > Thank you for your reply. We appreciate and respect your assessment of our work. But to be crystal clear, we would like to use this opportunity to clarify any possible misunderstanding. Please, see below our point-by-point response. (We rearrange the order of the quotes for clarity.)
> > >
> > > &ensp;
> > >
> > > > _Double descent, however, displays a decrease/increase in the model's performance (whose analogous would be I(T; W)) as the model reaches $N/P = 1$; and not of model relative performance vs a theoretical maximum (whose analogous would be I(T; W)/I(T; W))._
> > >
> > > We stress that the encoded relevant information I(T;W) is _not_ an analog of models' performance: models that memorize training data have maximum I(T;W) but they need not generalize well.
> > >
> > > The residual information I(T;Y|W) _alone_ is _not_ a proxy of models' performance either: models with random parameters have zero residual information but are completely uninformative of what we want to learn.
> > >
> > > An information-theoretic characterization of models' performance depends on _both_ I(T;W) and I(T;Y|W): good models encode relevant information without being too dependent on noise in the data.
> > >
> > > Placing a constraint on I(T;W) (e.g., by fixing $\mu$) allows us to use I(T;Y|W)&mdash;which measures how much a model depends on noise&mdash;as a performance proxy and focus on its sample-size dependence. (Alternatively we could make an assumption on I(T;Y|W) and use I(T;W) as a performance proxy.)
> > >
> > > &ensp;
> > >
> > > > _For each point in these curves, though, I(Y; W) changes---and thus so does I(T; W)._
> > >
> > > This statement is _not_ an artifact of our work. It is a feature of a _standard_ sample-wise double descent curve (generalization error vs sample size, see, e.g., the references below). More samples mean more available information about what we want to learn (relevant information); therefore, the models, fitted to more samples, necessarily encode more relevant information.
> > >
> > > - d'Ascoli, Sagun & Biroli, Triple descent and the two kinds of overfitting: where & why do they appear?, NeurIPS (2020)
> > > - Nakkiran, Kaplun, Bansal, Yang, Barak & Sutskever, Deep Double Descent: Where Bigger Models and More Data Hurt, ICLR (2020)
> > >
> > > &ensp;
> > >
> > > > _Namely, the figures show a "double descent" analog while holding $\mu=I(T; W)/I(Y; W)$ fixed._
> > >
> > > The nonmonotonic behaviors in Fig 2d-e are robust and _not_ a result of fixing $\mu$.
> > >
> > > To depict I(T;Y|W) (a measure of overfitting) as a function of sample size N, we must know how I(T;W) depends on N. We know that I(Y;W) increases with N so I(T;W) should also be an increasing function of N.
> > >
> > > Panel b of our supplementary figure (please, see the newly uploaded Supplementary Material) implies that I(T;Y|W) remains nonmonotonic in N as long as I(T;W) strictly increases with N. Holding $\mu$ fixed is equivalent to assuming that I(T;W) depends on N the same way that I(Y;W) depends on N, bar some constant multiplicative factor.
> > >
> > > It is only when we insist that I(T;W) is fixed (which is unrealistic given the available information goes up with N) that I(T;Y|W) becomes monotonic in N.
> > >
> > > &ensp;
> > >
> > > ---
> > >
> > > &ensp;
> > >
> > > Finally we would like to briefly summarize the main contribution of our work (repeating parts of our response to Reviewer Lvpt)
> > >
> > > Overfitting of course describes scenarios in which fitted models depend too much on noise in the training data, resulting in suboptimal generalization, but how do we quantify this dependence? And how much is too much? Our work offers an attractive answer to these two questions.
> > >
> > > First, we use mutual information as a measure of statistical dependence between random variables of interest. In particular, the residual information, I(T;Y|W)=I(T;Y)-I(T;W), quantifies how much fitted models depend on noise (the bits that are not informative of the generative process). While I(T;Y|W) can be related to the generalization gap, mutual information is more general than a particular loss function or performance metric.
> > >
> > > Second, we use the IB method to derive the minimum achievable residual bits; all learning algorithms must encode at least this much information about noise. This minimum value provides a baseline for how much fitted models _have to_ depend on noise. The information efficiency in Eq (11) offers an intuitive measure of how far a fitted model is from this baseline.
> > >
> > > Although generalization gaps somewhat measure how much models depend on noise, they are based on problem-specific performance metrics. In contrast, relevant and residual information and the IB principle are more general, and therefore our conceptual framework offers a step towards a unifying understanding of overfitting.
> > >
> > > To sum up, we formulate a new setting in which we can interrogate learning problems and uncover and interpret new interesting phenomenology from a new perspective. In our view, understanding the phenomenology of high-dimensional learning from different perspectives is key to unlocking new insights. We believe our work offers a step in this direction.

---

> > > > ### Comment · Reviewer_jTSy · 2022-08-09
> > > > **Reviewer Re-response**
> > > >
> > > > I thank the authors for their renewed comments.
> > > >
> > > > > We stress that the encoded relevant information I(T;W) is not an analog of models' performance: models that memorize training data have maximum I(T;W) but they need not generalize well.
> > > > >
> > > > > The residual information I(T;Y|W) alone is not a proxy of models' performance either: models with random parameters have zero residual information but are completely uninformative of what we want to learn.
> > > > >
> > > > > An information-theoretic characterization of models' performance depends on both I(T;W) and I(T;Y|W): good models encode relevant information without being too dependent on noise in the data.
> > > >
> > > > I agree with the authors that both I(T;W) and I(T;Y|W) determine performance together---and not only I(T; W). This is what I meant with my previous comment (although, while re-reading, I do see that I did not phrase my comment optimally).
> > > >
> > > > What i meant is that double's descent analogous would be to see a reduction in residual information $I(T;Y|W)$ at $\frac{N}{P}=1$ while holding $I(T;W)$ fixed; and not  while holding $\frac{I(T;W)}{I(Y;W)}$ fixed.
> > > >
> > > > > Placing a constraint on I(T;W) (e.g., by fixing ) allows us to use I(T;Y|W)—which measures how much a model depends on noise—as a performance proxy and focus on its sample-size dependence. (Alternatively we could make an assumption on I(T;Y|W) and use I(T;W) as a performance proxy.)
> > > >
> > > > This is exactly the kind of experiment that I would like to see. Holding $I(T;W)$ (but not $\frac{I(T;W)}{I(Y;W)}$) fixed and observing how $\frac{I(T;Y|W)}{I(\hat{T};Y|W)}$ changes as a function of it.
> > > >
> > > > I think the authors' parenthetical suggestion would also be a great extra experiment. Holding $\frac{I(T;Y|W)}{I(\hat{T};Y|W)}$ fixed and seeing how $I(T;W)$ changes as a function of it.
> > > >
> > > > > It is only when we insist that I(T;W) is fixed (which is unrealistic given the available information goes up with N) that I(T;Y|W) becomes monotonic in N.
> > > >
> > > > I'll expand now on why I think $I(T;W)$ (and not $\frac{I(T;W)}{I(Y;W)}$) should be held fixed in the experiments.
> > > > In short, this is because the uncertainty about W (i.e., $H(W)$) is held fixed in all cases---it does not depend on the sample size N.
> > > > Even though there is an upperbound to $I(T;W)$, i.e., $I(T;W) \leq I(Y;W)$, this is true also for double's descent observed accuracy results---i.e., there is a maximum expected accuracy you can achieve with linear regression when given a sample size N.

---

> > > > > ### Author Response · Authors · 2022-08-09
> > > > > **Author Response**
> > > > >
> > > > > Thank you for the clarification. We appreciate your time and effort in helping us improve our work. We have carefully considered and addressed all of your comments below. We hope our response helps clear up any possible misunderstanding. Please, let us know if we misunderstood any of your comments. (The following quotes are reordered for clarity.)
> > > > >
> > > > > &ensp;
> > > > >
> > > > > >_This is exactly the kind of experiment that I would like to see. Holding $I(T;W)$ (but not $\frac{I(T;W)}{I(Y;W)}$) fixed and observing how $\frac{I(T;Y|W)}{I(\hat{T};Y|W)}$ changes as a function of it._
> > > > >
> > > > > This is exactly what we depict in Supplementary Figure. Specifically, Panel c shows the information efficiency $\eta=I(\tilde T;Y|W)/I(T;Y|W)$ as a function of N/P (x-axis) and I(T;W) (y-axis). By taking a horizontal slice of this plot, we see clearly that, at fixed I(T;W), $\eta$ is nonmonotonic in N/P and is minimum around N/P=1.
> > > > >
> > > > > &ensp;
> > > > >
> > > > > >_I think the authors' parenthetical suggestion would also be a great extra experiment. Holding $\frac{I(T;Y|W)}{I(\hat{T};Y|W)}$ fixed and seeing how $I(T;W)$ changes as a function of it._
> > > > >
> > > > > We display this in Panel c of Supplementary Figure; the contour lines show how I(T;W) depend on N/P at a fixed information efficiency $\eta=I(\tilde T;Y|W)/I(T;Y|W)$. We see a clear nonmonotonic behavior. It takes more encoded relevant bits I(T;W) to achieve a prescribed efficiency as N/P approaches 1.
> > > > >
> > > > > &ensp;
> > > > >
> > > > > >_What i meant is that double's descent analogous would be to see a reduction in residual information $I(T;Y|W)$ at $\frac{N}{P}=1$ while holding $I(T;W)$ fixed; and not while holding $\frac{I(T;W)}{I(Y;W)}$ fixed._
> > > > >
> > > > > We emphasize that __none__ of the investigations of double descent constrains how much the models on double descent curves should know about the generative process. In other words, a _standard_ double descent curve does not have fixed I(T;W). In fact, in the case of sample-wise double descent (generalization error vs sample size), I(T;W) increases with sample size: models, fitted to more samples, encode more relevant information.
> > > > >
> > > > > We can of course characterize how I(T;Y|W) depends on N/P at fixed I(T;W) (e.g., by taking horizontal slices of Supplementary Figure), but this would be further from standard sample-wise double descent than allowing I(T;W) to increase with N.
> > > > >
> > > > > &ensp;
> > > > >
> > > > > >_I'll expand now on why I think $I(T;W)$ (and not $\frac{I(T;W)}{I(Y;W)}$) should be held fixed in the experiments. In short, this is because the uncertainty about W (i.e., $H(W)$) is held fixed in all cases---it does not depend on the sample size N. Even though there is an upperbound to $I(T;W)$, i.e., $I(T;W) \leq I(Y;W)$, this is true also for double's descent observed accuracy results---i.e., there is a maximum expected accuracy you can achieve with linear regression when given a sample size N._
> > > > >
> > > > > The fact that the differential entropy h(W) is independent of sample size N does not justify why double descent curves should have fixed relevant information I(T;W). Please, see our response to the previous comment on why letting I(T;W) vary with N is more natural for a double descent analog.
> > > > >
> > > > > The data processing inequality, I(T;W) <= I(Y;W), implies that if we fix I(T;W), then the sample size N must be greater than some finite value. This fact is at odds with _standard_ sample-wise double descent curves which do not place a lower bound on sample size. Indeed this is yet another reason why fixing I(T;W) is not an intuitive analog of sample-wise double descent.
> > > > >
> > > > > &ensp;
> > > > >
> > > > > Finally, we would like to thank the Reviewer again for engaging with us on these points. We will include a discussion of these subtleties in the final version of the paper.

---

> ### Author Response · Authors · 2022-08-02
> **Author Response (2/2)**
>
> _continued from previous comment_
>
> &ensp;
>
> >_If I understood correctly, in all plots where the x-axis is N/P only the N changes, right? Why display it as N/P instead of N then?_
>
> The quantities in our figures depend on the ratio N/P. But, for clarity, we choose to interpret the results from the perspective that P is held fixed (Sec 4, line 179). We feel that this perspective provides an intuitive setting in which the number of features (P) are fixed but the sample size (N) can vary. We will make this point clear in the figure captions of the updated paper.
>
> >_Figure captions say "Here we set [...] and let P, N -> \infty." What does it mean to let P, N -> \infty, while only N/P changes in the x-axis?_
>
> Thank you for pointing out another place where we can improve the clarity of our work. We send P and N to infinity at the same rate such that the ratio N/P remains fixed and finite (Sec 4, line 171). We will clarify this in the revised figure captions.
>
> >_Nitpick: Equation 1 discusses random variables S, T, and W . Why use S instead of Y?_
>
> We use S for training data in a general sense, and specialize to Y for the training data when specifically studying regression problems.

---

### Official Review · Reviewer_L3Rw · 2022-07-12

**Rating:** 5
**Confidence:** 3
**Soundness:** 3 good
**Presentation:** 3 good
**Contribution:** 2 fair

**Summary:**

This paper studied the over-fitting problem in linear models from an information-theoretic perspective. More specifically, the author defined the concept of information efficiency, which is the ratio between the mutual information values of the optimal representation and the learned representation to the scalar response. To further analyse the behavior of the information efficiency, the author considered the ridge regression with the Gibbs posterior, and derived the limit of the information efficiency with the relevance level going to one and temperature going to zero. Besides, the author checked the high dimensional limits of information under isotropic and anisotropic covariates setups.

**Questions:**

1. Is there any constraint on the optimal representation in equation (6) and (7)?

2. I am confused about the author's definition of information bottleneck with linear map. In my understanding, the information bottleneck method would minimize the mutual information between input X and representation T as I(T; X) , where the author minimized the mutual information between mapping W and the representation T as I(W;T). Could the author provide more explanation about this?

**Limitations:**

The derivation of the proposed theorem is under the simplest linear model, with some assumptions, which limits the generalization ability of the method for analyzing more complicated scenarios.

**Strengths And Weaknesses:**

Strength:
1. Different from the common usage of information bottleneck, which always served as a regularizer in learning processes, the author quantitatively analyzed the information efficiency of bottleneck representation with asymptopic limits.

2. The author observed the behaviors of mutual information from multiple perspectives such as efficiency, extensivity and redundancy.

3. The author smartly used the Gibbs posterior with zero-temperature limit to avoid the deterministic ridge regression.

Weakness:
1. The author's analysis is based on the simple linear model problem, which is too simple to apply the information bottleneck theory(since one can directly obtain the closed-form solution). Besides, I am wondering how the derived theorem can be further used into more complex practical scenarios, e.g. with nonlinearly neural networks.

2. The analysis of high dimensional limit with anisotropic covariates only consider the spectral distribution as a mixture of two-point masses. What if one considers more general scenarios?

---

> ### Author Response · Authors · 2022-08-02
> **Author Response**
>
> Thank you for highlighting the novelty of our study! We appreciate your comments and suggestions which have helped us improve our work.
>
> >__Strengths And Weaknesses:__
>
> >_The author's analysis is based on the simple linear model problem, which is too simple to apply the information bottleneck theory(since one can directly obtain the closed-form solution)._
>
> We note that "the closed-form solution" is a result of the IB theory. While it might look simple, its derivation is far from trivial (Chechik et al., 2005).
>
> >_Besides, I am wondering how the derived theorem can be further used into more complex practical scenarios, e.g. with nonlinearly neural networks._
>
> We agree that linear models are simple. However, they capture many important features of high-dimensional learning, of which our understanding is lacking, and can serve as a useful (if simple) setting for theoretical investigations. In fact, high-dimensional linear regression is far from being well-understood; it still generates new surprising results (Hastie et al., 2022) and provides much needed insights into high-dimensional learning, see, e.g., Bartlett et al. (2020), Wu & Xu (2020), Richards et al. (2021) and Mel & Ganguli (2021). In addition, linear models are instrumental in building our understanding of deep networks, see, e.g., Saxe et al. (2014), Lampinen & Ganguli (2019), Ji & Telgarsky (2019) and Arora et al. (2019).
>
> While we focus on linear regression in this work, our framework and the concepts we introduce apply to more complicated learning scenarios. One possible generalization would be to consider random feature models (Rahimi & Recht, 2007) for which available technical tools and results should make information-theoretic analyses tractable, see, e.g., Mei & Montanari (2019) and Mel & Pennington (2022) and references therein.
>
> >_The analysis of high dimensional limit with anisotropic covariates only consider the spectral distribution as a mixture of two-point masses. What if one considers more general scenarios?_
>
> Our formulation is readily applicable to arbitrary population spectral distributions $F^\Sigma$. However, we focus on the case of an equal mixture of two point masses to cleanly illustrate the effects of anisotropy. More spectral components would lead to more peaks in residual information, similarly to what we see in Fig 3d.
>
> >__Questions:__
>
> >_Is there any constraint on the optimal representation in equation (6) and (7)?_
>
> The only constraint is that the arguments of logarithms are nonnegative. This condition is always satisfied in our setup. The data processing inequality guarantees that $\gamma > 1$ [see, e.g., Wu & Fischer (2020)], and $0 \le \nu_i \le 1$ since the eigenvalues of the normalized regression matrix always range from 0 to 1 (Chechik et al., 2005). We will include this statement in our revised manuscript to avoid any confusion.
>
> >_I am confused about the author's definition of information bottleneck with linear map. In my understanding, the information bottleneck method would minimize the mutual information between input X and representation T as I(T; X) , where the author minimized the mutual information between mapping W and the representation T as I(W;T). Could the author provide more explanation about this?_
>
> Your understanding is correct and is consistent with the optimization in our work: we __maximize__ the relevant bits I(T;W) while minimizing the residual bits I(T;S|W), see Eq (1). Indeed, Eq (1) is identical to the original IB problem (Tishby et al., 1999) since I(T;S|W)=I(T;S)-I(T;W) under the Markov constraint T-S-W (see footnote 1). In the updated manuscript, we emphasize that the coefficient in front of I(T;W) in Eq (1) is negative and, as a result, minimizing Eq (1) is the same as maximizing I(T;W). Thank you for spotting this potential point of confusion!
>
> >__Limitations:__
>
> >_The derivation of the proposed theorem is under the simplest linear model, with some assumptions, which limits the generalization ability of the method for analyzing more complicated scenarios._
>
> Please, see our response above regarding linear models. We would like to further emphasize that our main contribution is the formulation of a new and informative framework for investigating learning problems. In particular, we demonstrate how the concepts from IB can be used to analyze learning algorithms, and we identify and interpret new information-theoretic phenomenology. Linear regression is very well suited for our purposes. In fact, we think of the theoretical tractability of linear settings as a feature that allows us to introduce new concepts, apply them to practical algorithms and discuss the many facets of high-dimensional learning (including overparametrization, regularization, double descent and covariance anisotropy) through the lens of information theory.

---

> ### Author Response · Authors · 2022-08-02
> **Author Response - References**
>
> __References:__ (in order of appearance)
>
> - Chechik, Globerson, Tishby & Weiss, Information bottleneck for Gaussian variables, JMLR 6, 165 (2005).
> - Hastie, Montanari, Rosset & Tibshirani, Surprises in high-dimensional ridgeless least squares interpolation, The Annals of Statistics 50, 949 (2022)
> - Bartlett, Long, Lugosi & Tsigler, Benign overfitting in linear regression, PNAS 117, 30063 (2020)
> - Wu & Xu, On the Optimal Weighted l2 Regularization in Overparameterized Linear Regression, NeurIPS (2020)
> - Richards, Mourtada & Rosasco, Asymptotics of Ridge(less) Regression under General Source Condition, AISTATS (2021)
> - Mel & Ganguli, A theory of high dimensional regression with arbitrary correlations between input features and target functions: sample complexity, multiple descent curves and a hierarchy of phase transitions, ICML (2021)
> - Saxe, McClelland & Ganguli, Exact solutions to the nonlinear dynamics of learning in deep linear neural networks, ICLR (2014)
> - Lampinen & Ganguli, An analytic theory of generalization dynamics and transfer learning in deep linear networks, ICLR (2019)
> - Ji & Telgarsky, Gradient descent aligns the layers of deep linear networks, ICLR (2019)
> - Arora, Cohen, Golowich & Hu, A Convergence Analysis of Gradient Descent for Deep Linear Neural Networks, ICLR (2019)
> - Rahimi & Recht, Random Features for Large-Scale Kernel Machines, NeurIPS (2007)
> - Mei & Montanari, The generalization error of random features regression: Precise asymptotics and double descent curve, arXiv:1908.05355
> - Mel & Pennington, Anisotropic Random Feature Regression in High Dimensions, ICLR (2022)
> - Wu & Fischer, Phase Transitions for the Information Bottleneck in Representation Learning, ICLR (2020).
> - Tishby, Pereira & Bialek, The information bottleneck method, 37th Allerton Conference on Communication, Control and Computing (1999)

---

### Meta-Review · Area_Chair_on1T · 2022-08-23

**Recommendation:** Accept
**Confidence:** Less certain

**Metareview:**

The technical concerns of the reviewers were cleared in the discussion and based on the final reviews and my own reading the paper seems technically sound. While the considered model is very simple and hence relevance to practice is hard to foresee, the contribution of analyzing the IB method in a simple setting is considered valuable and of interest to the community. We hence recommend acceptance of the paper. The reviews and the subsequent discussion provide many suggestions that should help the authors to improve the presentation of their results. In particular, adding the motivation for the information efficiency and plots suggested by reviewer jTSy seem of interest.

**Award:**

No

---

### Decision · Program_Chairs · 2022-09-14

Accept